# Characterization of Immunoactive and Immunotolerant CD4+ T Cells in Breast Cancer by Measuring Activity of Signaling Pathways That Determine Immune Cell Function

**DOI:** 10.3390/cancers14030490

**Published:** 2022-01-19

**Authors:** Yvonne Wesseling-Rozendaal, Arie van Doorn, Karen Willard-Gallo, Anja van de Stolpe

**Affiliations:** 1Molecular Pathway Diagnostics, Philips, 5656 AE Eindhoven, The Netherlands; yvonne.wesseling-rozendaal@philips.com; 2Philips Research, 5656 AE Eindhoven, The Netherlands; a.r.van.doorn@philips.com; 3Molecular Immunology Unit, Institut Jules Bordet, Université Libre de Bruxelles, 1000 Brussels, Belgium; karen.willard-gallo@bordet.be

**Keywords:** signal transduction pathway activity, breast cancer, immuno-oncology, immune cell types, immune tolerance, microarray

## Abstract

**Simple Summary:**

Immunotherapy enhances the immune response against cancer and is potentially curative. Unfortunately, few patients with breast cancer benefit from this therapy. It is not possible to predict which patients will benefit. A blood cell, called CD4+ T-cell, plays a role in the immune response and in resistance to immunotherapy. Its function is determined by activity of biochemical processes, called signal transduction pathways (STPs). We developed a new technology to measure activity of these STPs, which was used to investigate whether CD4+ T cells function abnormally in breast cancer patients. We show that in CD4+ T-cells from most of the investigated breast cancer patients a number of these STPs are overactive. The abnormal activity of a few notable STPs (Notch and TGFβ) suggests that CD4+ T-cells have changed into regulatory T-cells, which inhibit the immune response against cancer and have been associated with resistance to immunotherapy. We also provide evidence that this change in the CD4+ T- cells is caused by a factor produced by breast cancer cells. We conclude that this new technology can be used to measure STP activity in blood of patients with cancer and has the potential to better identify patients who will benefit from immunotherapy.

**Abstract:**

Cancer immunotolerance may be reversed by checkpoint inhibitor immunotherapy; however, only a subset of patients responds to immunotherapy. The prediction of clinical response in the individual patient remains a challenge. CD4+ T cells play a role in activating adaptive immune responses against cancer, while the conversion to immunosuppression is mainly caused by CD4+ regulatory T cell (Treg) cells. Signal transduction pathways (STPs) control the main functions of immune cells. A novel previously described assay technology enables the quantitative measurement of activity of multiple STPs in individual cell and tissue samples. The activities of the TGFβ, NFκB, PI3K-FOXO, JAK-STAT1/2, JAK-STAT3, and Notch STPs were measured in CD4+ T cell subsets and used to investigate cellular mechanisms underlying breast cancer-induced immunotolerance. Methods: STP activity scores were measured on Affymetrix expression microarray data of the following: (1) resting and immune-activated CD4+ T cells; (2) CD4+ T-helper 1 (Th1) and T-helper 2 (Th2) cells; (3) CD4+ Treg cells; (4) immune-activated CD4+ T cells incubated with breast cancer tissue supernatants; and (5) CD4+ T cells from blood, lymph nodes, and cancer tissue of 10 primary breast cancer patients. Results: CD4+ T cell activation induced PI3K, NFκB, JAK-STAT1/2, and JAK-STAT3 STP activities. Th1, Th2, and Treg cells each showed a typical pathway activity profile. The incubation of activated CD4+ T cells with cancer supernatants reduced the PI3K, NFκB, and JAK-STAT3 pathway activities and increased the TGFβ pathway activity, characteristic of an immunotolerant state. Immunosuppressive Treg cells were characterized by high NFκB, JAK-STAT3, TGFβ, and Notch pathway activity scores. An immunotolerant pathway activity profile was identified in CD4+ T cells from tumor infiltrate and blood of a subset of primary breast cancer patients, which was most similar to the pathway activity profile in immunosuppressive Treg cells. Conclusion: Signaling pathway assays can be used to quantitatively measure the functional immune response state of lymphocyte subsets in vitro and in vivo. Clinical results suggest that, in primary breast cancer, the adaptive immune response of CD4+ T cells may be frequently replaced by immunosuppressive Treg cells, potentially causing resistance to checkpoint inhibition. In vitro study results suggest that this is mediated by soluble factors from cancer tissue. Signaling pathway activity analysis on TIL and/or blood samples may improve response prediction and monitoring response to checkpoint inhibitors and may provide new therapeutic targets (e.g., the Notch pathway) to reduce resistance to immunotherapy.

## 1. Introduction

The infiltration of cancer tissue by a variety of immune cells is necessary to mount an adequate immune response against cancer cells. During the past decades, evidence has been accumulating that cancer tissue can be highly successful in creating an immune-tolerant environment, which interferes with the appropriate anti-cancer immune response of tumor-infiltrating (TIL) T cells. Checkpoint inhibitor immunotherapy against cytotoxic T lymphocyte antigen-4 (CTLA-4) and programmed death-1 (PD-1) aims at restoring the effector function of CD8+ cytotoxic T cells and when successful, has been shown to have the potential of being a curative treatment [1]. An increasing number of immunotherapy drugs that aim at re-activating the immune response in such a targeted manner is in development. Unfortunately, only a limited percentage of patients respond to this type of immunotherapy [2]. Assays to predict response, such as PD-1/PD-L1 and CD4+/CD8+ immunohistochemistry (IHC) staining measurements, have proven to be not sufficiently reliable in predicting response in the individual patient. Consequently, there is a high need for assays to predict therapy response and to assess, as soon as possible, whether the installed therapy is effective [2,3,4,5].

In the tumor infiltrate, CD4+ T cells play an important role in activating adaptive immune responses, and when reverted to an immune-suppressed state, they impair the anti-cancer immune response. As such, they present a potential therapeutic target to reverse an immunotolerant state. Indeed, a prognostic role for cancer-infiltrating CD4+ T cells with a regulatory T (Treg) cell phenotype has been described [6]. Such CD4+ Treg cells may play an immunosuppressive role, depending on their functional activation states. However, while specific gene expression was found to be associated with Treg infiltration in breast cancer tissue, the activation state of such Treg cells was not defined [7]. The functional state of CD4+ T cell types is regulated by signal transduction pathways, e.g., the PI3K, JAK-STAT1/2, JAK-STAT3, NFκB, TGFβ, and Notch pathways [8,9,10,11,12,13]. Recently, assays have been developed which enable the quantitative measurement of the activity of these signal transduction pathways in tissue and blood samples [14,15,16,17,18].

These signaling pathway assays are used to in vitro characterize resting, immune-activated and immunotolerant CD4+ T cells, as well as CD4+ T-helper 1 (Th1), T-helper 2 (Th2), and Treg cells, in terms of the activity of signaling pathways. We showed that the identified pathway activity profiles can be of help in elucidating the mechanism leading to cancer-induced immunotolerance and to investigate the type and the functional state of CD4+ T cells in blood, lymph nodes, and TIL samples from cancer patients [19].

## 2. Materials and Methods

### 2.1. Signaling Pathway Activity Analysis of Preclinical and Clinical Studies

Tests to quantitatively measure the functional activities of PI3K, NFκB, TGFβ, JAK-STAT, and Notch signal transduction pathways on Affymetrix Human Genome U133 Plus 2.0 expression microarrays data have been described before [15,16,18,20,21]. In brief, the pathway assays are based on the concept of a Bayesian network computational model which calculates, from mRNA levels of a selected set, usually between 20 and 30, target genes of the pathway-associated transcription factor, a probability score for the pathway activity, which is translated to a log2 value of the transcription factor odds, i.e., a log2 odds score scaling with the pathway activity. The models were all calibrated on a single-cell type and were subsequently frozen and validated on a number of other cell and tissue types without further adaptations of the models. The range (minimum–maximum pathway activities) on the log2 odds scale was different for each signaling pathway. While the signaling pathway assays can be used on all cell types, the log2 odds score range may vary per cell/tissue type. Of note, the measurement of the activity of the PI3K pathway assay was based on the inverse inference of the activity of the PI3K pathway from the measured activity of the FOXO transcription factor, in the absence of cellular oxidative stress [15]. For this reason, the FOXO activity score is presented in the figures, instead of the PI3K pathway activity. In cell culture experiments in vitro, the PI3K pathway activity can be directly (inversely) inferred from the FOXO activity. Activity scores were calculated for the FOXO transcription factor and NFκB, JAK-STAT, Notch, and TGFβ signaling pathways on Affymetrix expression microarray datasets from the Gene Expression Omnibus (GEO, www.ncbi.nlm.nih.gov/geo (accessed on 24 June 2020) database [22] or generated in our own lab and presented on a log2 odds scale [14,15,16].

The GSE71566 dataset contained Affymetrix data from CD4+ T cells isolated from cord blood. Naive CD4+ T cells were compared with CD4+ T cells, which were activated by treatment with anti-CD3 and anti-CD28, and with CD4+ T cells, which were differentiated to either Th1 or Th2 cells [23]. The GSE11292 dataset contained data from cell-sorted Treg cells (CD4+CD25hi), activated in vitro with anti-CD3/CD28 in combination with interleukin 2 (IL-2) in a time series [24].

Two Affymetrix datasets were generated in our own lab. They have been described in detail before [19] and are also publically available under accession numbers of GSE36765 and GSE36766. The GSE36766 dataset contains Affymetrix data from an in vitro study in which breast cancer tissue sections from fresh surgical specimens of primary untreated breast cancers (n = 4) were mechanically dissociated in X-VIVO 20 medium; CD4+ T cells from healthy donor blood were incubated for 24 h with anti-CD3/CD28 with or without the primary tumor X-VIVO 20 supernatant (SN), and microarray analysis was performed. The GSE36765 dataset is a clinical dataset containing data from patients with primary untreated breast cancer; methods and patient characteristics have been described by us in detail before [19]. In brief, CD4+ T cells were isolated from primary tumors, axillary lymph nodes, and peripheral blood of 10 patients with invasive breast carcinomas, as well as the peripheral blood of four healthy donors.

### 2.2. General Rules for the Interpretation of the Signal Transduction Pathway Activity Scores

An important and unique advantage of the pathway activity assays is that they can in principle be performed on each cell type. Important considerations for the interpretation of the log2 odds pathway activity scores are as following:(1)On the same sample, the log2 odds pathway activity scores cannot be compared between different signaling pathways, since each of the signaling pathways has its own range in the log2 odds activity scores;(2)The log2 odds range for the pathway activity (minimum–maximum activities) may vary, depending on the cell type. Once the range is defined using samples with a known pathway activity, on every new sample, the absolute value can be directly interpreted against that reference. If the range is not defined, only differences in the log2 odds activity score between samples can be interpreted;(3)The pathway activity scores are highly quantitative, and even small differences in log2 odds can be reproducible and meaningful;(4)A negative log2 odds ratio does not mean that the pathway is inactive.

### 2.3. Microarray Data Source and Quality Control

The analyzed datasets contained Affymetrix Human Genome U133 Plus 2.0 expression microarray data. Quality control (QC) was performed on the Affymetrix data of each individual sample based on 12 different quality parameters following Affymetrix recommendations and previously published literature [25,26]. In summary, these parameters include the average value of all probe intensities, presence of negative or extremely high (>16-bit) intensity values, poly-A RNA (sample preparation spike-ins) and labelled cRNA (hybridization spike ins) controls, *GAPDH* and *ACTB* 3′/5′ ratio, the center of intensity and values of positive and negative border controls determined by affyQCReport package in R, and an RNA degradation value determined by the AffyRNAdeg function from the Affymetrix package in R [27,28]. Samples that failed QC were removed prior to data analysis.

### 2.4. Statistics

Mann–Whitney U testing was used to compare the pathway activity scores across groups. In case another statistical method was more appropriate due to the content of a specific dataset, this is indicated in the legend of the figure. For pathway correlation statistics, Pearson correlation tests were performed. Exact *p*-values are indicated in the figures.

### 2.5. Defining a Threshold for the Abnormal Pathway Activity in Blood CD4+ T Cell Samples

To enable the clinical use of the pathway activity test on blood samples, a preliminary threshold value was calculated, above which the signaling pathway activity may be considered as abnormally high. The GSE36765 dataset contained blood-derived CD4+ T cell samples from four healthy individuals. For each pathway, the mean pathway activity score ± 2SD was calculated, and the upper threshold for the normal (in healthy individuals) pathway activity was defined as the mean ± 2SD.

## 3. Results

### 3.1. Measuring Activity of Signal Transduction Pathways, Using Validated Assays

All assays for quantitatively measuring signal transduction pathway activity that were used to obtain the results described below, which were validated before on multiple cell and tissue types, for references and guidance on the interpretation of the calculated signaling pathway activity scores (see Methods).

The activities of the FOXO transcription factor and of the NFκB, JAK-STAT1/2, JAK-STAT3, Notch, and TGFβ signaling pathways were measured in resting and activated CD4+ T cells, in CD4+ T cells differentiated to Th1 and Th2 cells, in Treg cells, in CD4+ T cells incubated with breast cancer SN, and in a series of matched blood, lymph nodes, and TIL samples from patients with breast cancer. As mentioned, pathway activity scores are presented on a log2 odds scale, with a varying activity range (from minimum to maximum measured pathway activities) on this scale per pathway and cell/tissue type.

### 3.2. Signaling Pathway Activities in Resting and Activated CD4+ T Cells, in CD4+-Derived Th1 and Th2 Cells, and in CD4+ Treg Cells

CD4+ T cells can have different phenotypes with specialized functions in the immune response, such as Th1, Th2, and strongly immune-suppressive Treg cells. We performed signaling pathway analysis on these different CD4+ T cell types generated in vitro and observed very specific signaling pathway activity profiles for the different cell types. In all samples, the PI3K pathway activity could be inversely inferred from the FOXO activity, since there was no evidence for cellular oxidative stress interfering with the inverse relationship between the FOXO and PI3K pathway activities (results not shown) [15]. The conventional activation of CD4+ T cells obtained from cord blood with anti-CD3/anti-CD28 antibodies resulted in the activation of the NFκB pathway and slightly increased JAK-STAT1/2 and JAK-STAT3 pathway activities, while the activity of the TGFβ pathway, already low, was further reduced (Table 1A). The FOXO activity decreased, indicating the activation of the PI3K growth factor pathway. Activated T cells which were further differentiated in vitro to Th1 and Th2 T cells showed differential changes in the pathway activity: NFκB and JAK-STAT1/2 pathway activities increased in Th1, while the activity of the FOXO transcription factor increased in Th2 cells (Table 1B).

In the activated CD4+ T cells from our own dataset, in which CD4+ T cells were derived from peripheral blood from healthy volunteers and subsequently were activated by anti-CD3/anti-CD28 incubation, a similar CD4+ T cell pathway activity profile was observed after activation with anti-CD3/CD28 (Table 2, comparing the non-activated control with the activated control) [19].

A distinct type of CD4+ T cells was the Treg cells, which in the analyzed set were activated with anti-CD3/CD28 plus IL-2. Treg cells were clearly distinguishable from the other CD4+ T cell types. Resting-state Treg cells mostly resembled Th2 cells, but with a slightly higher JAK-STAT1/2 pathway activity, a lower JAK-STAT3 pathway activity, and a higher Notch pathway activity (Table 1C). This dataset also included two experiments in which a Treg immune-suppressive state (induced Treg cells; iTreg) was induced and compared to activated T effector cells. Although the two experiments lacked good reproducibility and some sample data needed to be removed because of QC failure, the sampling at multiple time points after activation allowed a tentative interpretation. The activation of Treg cells towards iTreg was associated with the increased FOXO activity, reflecting a reduction in the PI3K pathway activity and increased NFκB, JAK-STAT3, and TGFβ pathway activities, and a reduced JAK-STAT1/2 pathway activity. The Notch pathway activity was higher in Treg cells compared to in T effector cells (both in the resting and activated states) and showed a consistent time-related increase in the pathway activity in one experiment (Table 1C, Appendix A).

These quantitative pathway analysis results supported a mechanistic role for the PI3K-FOXO, NFκB, JAK-STAT1/2, and JAK-STAT3 pathways during the activation of CD4+ T cells differentiation to Th1 and Th2 T cells, while the activities of the TGFβ and Notch signaling pathways were specifically linked to iTreg cells.

### 3.3. Immune-Suppressive Effect of the Cancer Cell SN on the Activated CD4+ T Cells

Our previously published study, in which we investigated the effect of adding a breast cancer tissue SN) to resting or activated CD4+ T cells of healthy individuals, allowed us to analyze the effect of breast cancer tissue on the signaling pathway activity in CD4+ T cells (Table 2) [19]. The flow cytometry analysis of resting and CD3+/CD28+-activated and SN-treated CD4+ T cells showed before that in these CD4+ T cells tumor, the SN inhibited the activation-induced expression of CD69, ICOS, and CD25 proteins (see Figure 6A,B in reference [19]). While the addition of SN to four different dissected primary breast cancer tissue samples did not affect signaling pathway activities in resting CD4+ T cells, the same SN changed pathway activity scores in activated CD4+ T cells: NFκB, and JAK-STAT1/2 and JAK-STAT3 pathway activity decreased, while the activity of the TGFβ pathway increased (Table 2). In addition, the FOXO activity increased, reflecting a decrease in the PI3K pathway activity as a consequence of incubation with the tumor SN. Importantly, after the addition of the SN, the JAK-STAT3 pathway activity remained higher than in resting CD4+ T cells, while the activity scores of the FOXO transcription factor increased even above those seen in resting CD4+ T cells. In summary, the cancer tissue SN caused a partial reversion of the pathway activity profile towards that of resting CD4+ T cells in combination with a strong upregulation of the immune-suppressive TGFβ pathway and a decrease in the PI3K pathway activity, in line with the induction of an immunotolerant state.

### 3.4. Clinical Study: Signaling Pathway Activity in CD4+ T Cells Derived from Blood, Lymph Nodes, and Breast Cancer Tissue Samples of Patients with Primary Breast Cancer

Having characterized the signaling pathway activity profiles associated with activated (anti-CD3/CD28) and immunotolerant CD4+ T cells in vitro, we proceeded to investigate CD4+ T cells that were obtained previously in a clinical patient setting [19]. The matched CD4+ T cells from peripheral blood, lymph nodes, and tumor infiltrate of 10 early untreated breast cancer patients were compared with respect to the above identified signaling pathway activities.

The Notch pathway activity was significantly increased (*p* = 0.002) in blood samples from breast cancer patients compared to from healthy individuals (Figure 1). Increased JAK-STAT3, and TGFβ pathway activity scores were measured in a sample subset, but compared to the four control samples, this did not reach a statistically significant difference (Figure 1). In CD4+ T cells isolated from TIL samples, NFκB, JAK-STAT1/2, JAK-STAT3, TGFβ, and Notch pathway activity scores were significantly increased compared to the pathway activity scores in the corresponding patient blood samples (Appendix A). The FOXO activity was also increased in TIL samples, indicative of an inactivated PI3K pathway (Figure 1 and Appendix A). Comparing the absolute pathway activity scores of TIL samples with the pathway scores measured in the in vitro cancer SN study, TIL-derived CD4+ T cells mostly resembled in vitro activated CD4+ T cells that were incubated with the cancer tissue SN (Table 2, Figure 1). To identify which subset of CD4+ T cells was responsible for the observed pathway profile in the TIL samples, this was compared with the in vitro identified profiles. The TIL-derived CD4+ T cells profile was in the majority of patients, which was highly similar to the iTreg cell profile, characterized by the low PI3K pathway activity and the high JAK-STAT3, NFκB, TGFβ, and Notch pathway activities.

The activities of the NFκB, JAK-STAT3, TGFβ, and Notch signaling pathways were strongly correlated in TIL samples, suggesting that they were often combined active in the same CD4+ T cells and that these CD4+ T cells were iTreg cells (Appendix A).

Compared to in the in vitro T cell experiments, in the clinical samples, pathway activity scores varied more between patients, reflecting the heterogeneity of the analyzed breast cancer patient group that consisted of four ER/PR-positive luminal patients, one ER/PR/HER2-positive patient, and five triple-negative patients (Table 3, Appendix A) [19]. Although the number of patients per subtype was too small to draw any conclusions on subtype specific pathway profiles in CD4+ TIL cells, it is interesting to note that the TGFβ and NFκB pathway activities were significantly lower in triple-negative patients compared to in the other subtypes. In contrast, in ER/PR-positive (HER2-negative) cancer samples, the immune-suppressive iTreg profile (high NFκB, JAK-STAT3, TGFβ, and Notch pathway activity scores) was most prominently present, compared to in the triple-negative cancer samples.

### 3.5. A Threshold for the Abnormal Pathway Activity in Blood CD4+ T Cell Samples

For each pathway, the mean pathway activity score ± 2SD was calculated, and the upper threshold for the normal pathway activity was defined as the mean ± 2SD (Appendix A). When applying these thresholds to the patient samples, a combined Notch, JAK-STAT3, and TGFβ pathway activity was identified in two patients, and the abnormal Notch pathway activity was identified in 9 out of 10 patients (Table 4 and Appendix A).

## 4. Discussion

A recently developed assay platform for measuring the activities of the most relevant signal transduction pathways was used to in vitro characterize the functional states of different subsets of CD4+ T cells with respect to the signaling pathway activity, identify the pathway profile of immunotolerant CD4+ T cells and analyze the immunosuppressive effect of the cancer tissue SN. Subsequently, a comparison was made with CD4+ T cell samples from breast cancer patients to identify the CD4+ T cell subset responsible for immunotolerance in TIL samples, and to investigate whether the immunotolerant CD4+ T cell pathway profile can also be detected in blood samples from breast cancer patients. The signal transduction pathway assays used in the current study have been biologically validated on multiple cell types, including immune cell types [14,15,16,18]. Affymetrix expression microarray data were used for the signaling pathway analysis; however, it is good to keep in mind that with this assay technology Affymetrix microarrays are only used as a method to measure a carefully preselected set of mRNA levels and not to discover a new gene profile. From Affymetrix whole transcriptome data, the expression data of only 20–30 pathway target genes per signaling pathway were used to calculate a signaling pathway activity score, and the respective genes have been described [14,15,16,18,20].

The measured changes in pathway activity scores associated with CD4+ T cell activation were consistent across two independent studies [19,23]. Increased PI3K (inferred from a decrease in the FOXO activity), NFκB, JAK-STAT1/2, and JAK-STAT3 pathway activities reflected T cell activation and are of crucial importance for the clonal proliferation and execution of specific immune response functions [8,9,10,29,30,31,32,33,34]. The differentiation of activated CD4+ T cells to Th1, Th2, and Treg cells appeared to be associated with characteristic alterations in the pathway activity profile, reflecting specific cell functions determined by the signaling pathway activity [15,16,18,21]. For example, in Th1 compared to in Th2 cells, a higher JAK-STAT1/2 pathway activity is necessary for the Th1 role to activate CD8+ T cells, for example in viral infections [9], while a high PI3K pathway activity is required for the expression of the TBET transcription factor, essential for Th1 function [8]. In immune-suppressive iTreg cells, a combined activity of Notch with JAK-STAT3 and TGFβ pathways is in line with these pathways controlling the immune suppressive function in a cooperative manner [35,36]. The very low PI3K pathway activity reflects the minimal role for this signaling pathway in Treg cells [8].

The cancer tissue SN appeared to induce a partial reversal of the CD4+ activation profile in vitro, resulting in a profile which was very similar to that described for immunotolerant lymphocytes and suggesting that one or more soluble factors from cancer tissue had induced immunotolerance [33,37,38]. An earlier analysis of this study using conventional bioinformatics approaches resulted in a similar conclusion as to a tumor suppressive effect of the cancer SN, but without linking this to the signaling pathway activity [19]. Potential candidates capable of inducing the identified pathway activity profile are a soluble form of PD-L1, TGFβ, and IL-6 [11,39,40,41,42,43,44,45]. The PD-L1 receptor PD-1 mediates tolerance induction in part by inhibiting activation of PI3K, explaining the observed inhibition of the PI3K pathway [38,46]. TGFβ is locally produced and can directly activate the tumor-suppressive TGFβ pathway [11,43,44]. Interleukin 6 (IL-6) induces the activation of the JAK-STAT3 pathway [45]. The crosstalk between these signaling pathways may be required to induce full T cell tolerance [47,48,49].

While the JAK-STAT3 pathway was found moderately activated in immunotolerant CD4+ T cells, higher pathway activation scores were found in properly activated CD4+ T cells, suggesting a dual function for this pathway. Indeed, the transcriptional program of the STAT3 transcription factor depends on the crosstalk with other signaling pathways in the cell, resulting in T cell activation in the presence of CD3/CD28 stimulation or in immunotolerance in the presence of immunosuppressive factors such as TGFβ, IL-6, and PD-L1 [9,11,29,37,43].

Following in vitro CD4+ T cell subset analysis, CD4+ T cells from blood, lymphocyte, and TIL samples from patients with untreated primary breast cancer were analyzed. The variation in the pathway activity scores between patients was larger in these clinical samples, probably caused by variable immune responses determined by factors such as the antigenicity of the tumor and genetic variations.

A subset of TIL-derived samples, mostly from luminal (ER/PR-positive) breast cancers, had a pathway activity profile resembling the immune-tolerant pathway profile found in the tumor SN-treated cells with respect to the PI3K, JAK-STAT3, and TGFβ pathway activities, complemented by higher Notch and NFκB pathway activities. The comparison of these TIL samples with Th and Treg subset pathway profiles revealed a strong match with iTreg cells, including the Treg-specific Notch pathway activity. Treg cells induce immune suppression in cancer tissue and are thought to mediate resistance against checkpoint inhibitors [50,51,52]. Indeed, patients with ER/PR-positive breast cancer are generally unresponsive to checkpoint inhibitor treatment [53]. The higher JAK-STAT1/2 and NFκB pathway activities in TIL compared to in vitro iTreg cells may be due to local inflammatory conditions in cancer tissue [54].

In patient blood CD4+ T cells, the Notch pathway activity was significantly increased compared to in healthy controls, similarly suggesting the presence of immune-suppressive Treg cells. The JAK-STAT3 and TGFβ pathway activities were also higher in patient blood samples, but they did not reach significance, which can be explained by a mixture of CD4+ T cell subsets being present in blood and a smaller fraction of activated Treg cells than in the TIL samples [50]. Using bioinformatics analysis, in the earlier reported analysis, no significant differences in the transcriptome were identified between CD4+ T cells from healthy and cancer patients [19]. Current findings may be explained by a Treg subset of CD4+ T cells having cycled from cancer tissue into blood; alternatively, elevated levels of circulating cytokines such as IL-6, IL-10, and TGFβ may explain the tumor-suppressive profile in these blood samples [50,55]. Our preferred hypothesis is that TGFβ plays a dominant role in the conversion of CD4+ T cells to iTreg cells, because TGFβ is known to be produced in cancer tissue, is associated with an immunosuppressive state and induces the expansion of Treg cells [44]. Thus, TGFβ is the most likely factor to explain both an immunotolerance-inducing effect of cancer SNs in vitro and an expanded Treg cell population in patient blood samples.

It is of potential clinical relevance that already in primary untreated breast cancer, patients activated immune-suppressive Treg cells may be present in peripheral blood. While Treg cells can be identified and isolated using flow cytometry and FACS techniques, our pathway analysis enabled the assessment of resting versus activated functional states. The first clinical study is in progress to investigate whether measuring this activated Treg profile in blood may help to predict resistance to checkpoint inhibitor therapy. For Affymetrix-based signaling pathway analysis, the defined (preliminary) thresholds for the abnormal pathway activity will be used; thresholds will be adapted for qPCR-based pathway activity profiling.

## 5. Limitations of the Study

A limitation of the current study is the small number of patients involved. Although results were already significant, future clinical studies should be directed at extending this cohort and provide further confirmation. In addition, limited data were available on the various types of in vitro CD4+ T cells, and their respective pathway activity profiles need future confirmation.

## 6. Conclusions

Using Philips signaling pathway analysis, we provided evidence that soluble factors in primary breast cancer tissue induce an immune-tolerant CD4+ T cell state, likely caused by activated immune-suppressive Treg cells that may already be detectable in blood samples. Since activated Treg cells may confer resistance to checkpoint inhibitors, the signaling pathway analysis of CD4+ T cells in TIL or in blood samples of cancer patients may enable an improved prediction and monitoring of response to checkpoint inhibitor therapy.

## Figures and Tables

**Figure 1 cancers-14-00490-f001:**
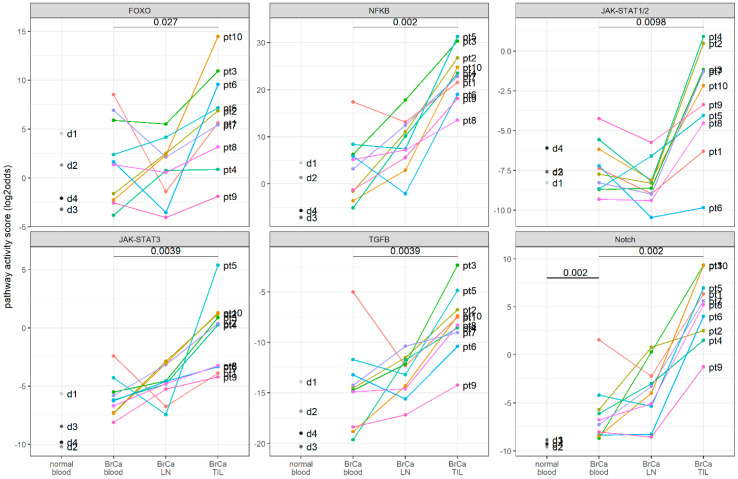
Signal transduction pathway activity in CD4+ T cells from matched blood, lymph nodes (LNs), and tumor-infiltrating (TIL) samples of 10 patients with invasive breast cancer (BrCa) and four healthy donors (dataset GSE36765) [19]. From left to right: FOXO, NFκB, JAK-STAT1/2, JAK-STAT3, TGFβ, and Notch pathway activity scores. The pathway activity is indicated as log2 odds on the *y*-axis. For each individual patient, the sample pathway scores are connected by colored lines (“pt” for BrCa patient samples; “d” for healthy donor samples). *p*-values (two-sided Wilcoxon-test) are listed when significant (*p* < 0.05); a paired test was performed when comparing results amongst BrCa samples.

**Table 1 cancers-14-00490-t001:** Signal transduction pathway activity in CD4+ T cells. (**A**,**B**) GSE71566, CD4+ T cells derived from cord blood, resting or activated using anti CD3/CD28; replicates indicate three cord blood samples (**A**) or stimulated with IL-12 and IL-4 for differentiation towards T-helper 1 (Th1) and T-helper 2 (Th2) T cells, respectively; replicates indicate three cell culture experiments (**B**) [23]. (**C**) GSE11292, CD4+ Treg cells, resting or activated with anti-CD3/-CD28 and IL-2, results from one Treg-activation experiment. For complete data analysis results, see Appendix A [24]. The first column contains the sample annotation as available from the GEO; the top of columns contains the names of signaling pathways; note that the FOXO transcription factor activity is presented, which is the inverse of the PI3K pathway activity. Signaling pathway activity scores are presented on a log2 odds scale with color coding ranging from blue (most inactive) to red (most active).

A.	Sample Category	Annotation Per Sample	FOXO	NFκB	JAK-STAT1/2	JAK-STAT3	TGFβ	Notch
	CD4+ naive	replicate 1	10.3	−9.6	−7.7	−7.3	−10.9	−5.0
		replicate 2	12.4	−8.2	−8.2	−8.1	−10.9	−3.4
		replicate 3	7.5	−11.7	−9.2	−6.3	−11.0	−4.0
	CD4+ activated	replicate 1	−3.8	−1.8	−2.8	−3.4	−18.8	−4.2
		replicate 2	−3.5	−1.7	−4.4	−2.5	−18.9	−3.9
		replicate 3	−4.9	−0.8	−2.5	−1.6	−18.2	−4.1
**B.**	**Sample Category**	**Annotation Per Sample**	**FOXO**	**NFκB**	**JAK-STAT1/2**	**JAK-STAT3**	**TGFβ**	**Notch**
	CD4+ activated + interleukin 12 (IL-12)	replicate 1	−4.9	5.3	−1.1	−0.1	−15.6	−4.5
		replicate 2	−4.4	6.1	−0.9	−1.1	−14.2	−4.6
		replicate 3	−4.8	5.2	−1.3	−0.3	−14.6	−3.9
	CD4+ activated + IL-4	replicate 1	0.6	−6.2	−7.5	−0.2	−19.0	−4.2
		replicate 2	−0.4	−5.1	−7.0	0.0	−18.3	−3.6
		replicate 3	0.2	−5.1	−5.9	0.6	−18.5	−4.0
**C.**	**Sample Category**	**Annotation Per Sample**	**FOXO**	**NFκB**	**JAK-STAT1/2**	**JAK-STAT3**	**TGFβ**	**Notch**
	Treg, in a resting state		−0.7	−6.7	−2.8	−2.5	−16.3	−0.6
	Treg, in a suppressive state	180 min	8.9	13.3	−5.1	17.0	−9.8	3.6
		200 min	10.6	12.4	−4.7	17.1	−8.9	3.3
		220 min	10.0	12.7	−5.2	16.6	−7.9	1.6

**Table 2 cancers-14-00490-t002:** Effect of the breast cancer tissue supernatant on the signal transduction pathway activity Scheme 4. T cells. According to dataset GSE36766, resting or in vitro activated (anti-CD3/CD28) CD4+ T cells from healthy donor blood were incubated with the primary cancer tissue supernatant (n = 4 different cancers) or control medium (n = 3) [19]. The left set of columns contained information on the experimental protocol for each sample. Signaling pathway activity scores are presented on a log2 odds scale with color coding ranging from blue (most inactive) to red (most active).

Sample Category	Annotation per Sample		FOXO	NFκB	JAK-STAT1/2	JAK-STAT3	TGFβ	Notch
Not activated	Control	1	−2.7	−10.9	−8.3	−6.2	−14.8	−7.5
		2	−2.4	−10.8	−8.8	−6.3	−14.3	−7.3
		3	−3.2	−11.1	−8.7	−6.8	−14.8	−7.1
	+ cancer tissue supernatant	tumor 1	−2.6	−9.1	−8.3	−7.5	−10.1	−6.8
		tumor 2	−2.4	−11.5	−8.4	−8.3	−10.3	−7.2
		tumor 3	−1.7	−9.2	−8.5	−5.5	−11.7	−7.4
		tumor 4	−2.0	−11.2	−8.7	−7.5	−13.2	−7.8
Activated (anti-CD3/anti-CD28)	Control	1	−7.5	3.5	−6.2	6.0	−15.8	−4.2
		2	−7.2	2.3	−5.9	5.2	−16.4	−4.3
		3	−7.5	3.1	−6.0	5.0	−17.0	−4.2
	+ cancer tissue supernatant	tumor 1	9.1	−10.0	−9.0	−2.1	−1.4	−7.3
		tumor 2	8.5	−10.0	−8.9	−1.9	−1.4	−5.8
		tumor 3	3.5	−4.4	−7.0	−1.1	−6.5	−4.8
		tumor 4	5.9	−5.9	−8.7	−1.3	−3.8	−4.3

**Table 3 cancers-14-00490-t003:** GSE36765. The pathway activity in TIL CD4+ T cells related to the additional clinical information which was described before [19]. Signaling pathway activity scores are presented as log2 odds with color coding ranging from blue (most inactive) to red (most active). The TGFβ and NFκB pathway activities were significantly lower in triple-negative patients compared to in the other subtypes (*p* = 0.028 and *p* = 0.028, respectively).

Annotation per Sample	FOXO	NFκB	JAK-STAT1/2	JAK-STAT3	TGFβ	Notch	Age	ER ^a^	PR ^b^	HER2 ^c^	Tumor Size (cm) ^d^	LN Metastasis ^e^	T cell Infiltrate ^f^	Histological Grade ^g^
TIL pt 3	10.9	30.3	−1.2	0.9	−2.4	9.3	43	+	+	−	2.8	0/12	minimal	3 (SBR8)
TIL pt 1	5.6	21.5	−6.3	−3.9	−7.5	6.3	51	+	+	−	1.8	0/16	minimal	2 (SBR6)
TIL pt 5	7.2	31.3	−4.0	5.4	−4.8	7.0	58	+	+	−	4	1/10	minimal	2 (SBR7)
TIL pt 2	6.9	26.8	0.5	1.2	−6.8	2.5	56	+	+	−	1.8	1/19	extensive	2 (SBR7)
TIL pt 4	0.9	23.5	0.9	0.3	−8.5	1.5	42	+	+	+	4.4	7/26	extensive	3 (SBR9)
TIL pt 10	14.5	24.8	−2.2	1.3	−7.4	9.3	61	−	−	−	3.5	0/11	extensive	3 (SBR9)
TIL pt 7	5.4	22.8	−1.3	0.4	−9.0	5.6	54	−	−	−	1.8	0/13	extensive	3 (SBR9)
TIL pt 6	9.6	19.0	−9.8	−3.3	−10.4	4.0	74	−	−	−	0.4 M	0/14	minimal	3 (SBR8)
TIL pt 8	3.2	13.6	−4.5	−3.2	−8.3	5.2	68	−	−	−	2.7	0/20	minimal	3 (SBR8)
TIL pt 9	−1.9	18.1	−3.4	−4.2	−14.2	−1.3	82	−	−	−	2.5 M	20/22	minimal	3 (SBR8)
^a^ Estrogen receptor 1 (*ESR1*); estrogen receptor alpha ^b^ Progesterone receptor. ^c^ CD340, HER-2/neu, *ERBB2*. ^d^ For multi-focal tumors, the size of the largest invasive foci is indicated followed by M for multifocal. ^e^ The number of positive LNs/number of examined LNs. ^f^ The extent of T cell infiltrate determined by immunohistochemistry (IHC; CD3 and CD4 labeling) ^g^ Scarff Bloom-Richardson staging score Breast cancer subtypes are color coded according to IHC staining:														
ER^+^ HER2^−^			HER2^+^			ER^−^ HER2^−^								

**Table 4 cancers-14-00490-t004:** Abnormal pathway activities in blood-derived CD4+ T cells from 10 patients with primary breast cancer (GSE36765) [19], using the normal pathway activity score (mean ± 2SD) as an upper threshold for the normal pathway activity. Pathway activities classified as abnormal are highlighted in gray.

	FOXO	NFκB	JAK-STAT1/2	JAK-STAT3	TGFβ	Notch
patient 1	8.5	17.4	−7.4	−2.4	−5.0	1.6
patient 2	−1.6	−1.5	−7.7	−7.3	−14.5	−5.7
patient 3	5.9	6.3	−8.7	−5.5	−14.7	−8.7
patient 4	−3.8	−5.1	−5.6	−6.2	−19.6	−6.1
patient 5	2.4	8.4	−8.7	−4.3	−11.7	−4.2
patient 6	1.7	5.9	−7.2	−6.3	−13.2	−8.4
patient 7	6.9	3.2	−8.3	−5.8	−14.3	−7.3
patient 8	1.3	5.2	−9.3	−6.7	−14.9	−6.8
patient 9	−2.5	−1.3	−4.2	−8.1	−18.4	−8.0
patient 10	−2.2	−3.6	−6.2	−7.3	−18.8	−8.4

## Data Availability

All analyzed microarray data are publicly archived datasets available from the GEO database (https://www.ncbi.nlm.nih.gov/gds/). Generated pathway activity scores are available in the Appendix A.

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
