# Peer review of "Characterization of Immunoactive and Immunotolerant CD4+ T Cells in Breast Cancer by Measuring Activity of Signaling Pathways That Determine Immune Cell Function"

_cancers, 2022, doi:10.3390/cancers14030490_

Round 1

Reviewer 1 Report

Although the authors made a few changes in the revised MS, as a matter of fact, the authors did not convincingly and satisfyingly address my previous major concerns without much improvement regarding those points in the revised MS. 

On the other hand, to make this MS more convicing and significant, the authors should at least include more experimental validation on the in vitro data (such as western blot, flow cytometry) to demonstrate those changes, although the authors added a few discussions regarding the limitations of this study in the revised MS.

Reviewer 2 Report

The authors have addressed concerns about the small sample size and included a discussion of the limitations of the study.

Author Response

Thanks for your valuable opinion.

Round 2

Reviewer 1 Report

Thanks for further addressing my concerns.

This manuscript is a resubmission of an earlier submission. The following is a list of the peer review reports and author responses from that submission.

Round 1

Reviewer 1 Report

In the manuscript entitled "Characterization of immunoactive and  immunotolerant CD4+ T cells in breast cancer by measuring activity of signaling pathways that determine immune cell function", the authors tried to check the status / changes of CD4+ T cells in breast cancer patients by using published microarray data. Through the study, they found CD4+ Th1, Th2, and Treg cells had specific Immune Pathway Activity Profiles (I-PAP) scores, and immunosuppressive Treg cells showed high NFκB, JAK-STAT3, TGFβ, and Notch pathway activity scores. Cancer supernatant reduced PI3K, NFκB, 23 and JAK-STAT3 pathway activity, but increased TGFβ pathway activity.  Further, the authors claimed CD4+ T cells from tumor infiltrate and blood of a subset of primary breast cancer patients showed immunotolerant PAP scores, which could be contributed to immunosuppressive Treg cells. There are some both major and minor concerns regarding this manuscript. 

Major:

1) There is low novelty of this study regarding both concept and study techniques. There have already studies have revealed the infiltration of Treg cells in breast cancers, the induction of immunosupression and its value for prognosis, such as PMID 27851913 and 33602289. Even this study did not have strong / direct evidence to further support its conclusion, just only from some signaling pathways. 

2) The presentaion of the data was bad with only some tables, which were even not well made. We can read the tables, but it is hard to understand what conclusion they want to make. It is better to add some graphs as well if they want to compare some groups. In this way, it will be much clear how each pathway change in cancer patients or different treatments. Those tables also occupied large space with lots of empty space. The authors should make / adjust the tables better and more readable. 

3) Some analysis is not professional and scientifically presented. For example, Figure 3 should be changed. I guess the authors wanted to show how the pathway/activity scores are from blood, LN and TIL of each patient, so they connected those scores from different tissues with lines. However, this is not correct. These lines can be drawn when you want to show the changes before / after treatment or how it changes as time goes, but not the case in this MS.

4) The method part was clear and not well described. Please explain the datasets clearly and it is better to use a table to summarize those datasets, including GEO number, experimental condition, in vitro cells or patient samples...

5) Again, the presentation of Table 1 is bad and should be changed.

Minor: 

1) What do replicate 1, 2, 3 exactly meant in the tables?

2) How did you get those numbers in the tables? why are most of them are negative even in the controls? 

3) Introduction / background is not enough, some key publications were not included.

4) Line 160, make the format of the words consistent.

5) Two "Supplementary Figure S3" labels in the supplement.

6) Line 232,  Should be "Supplementary Figure S3" other than  "Supplementary Figure 3". Make the format consistent in the MS and supplement.

Reviewer 2 Report

In this manuscript, the authors measured the I-PAP scores based on Affymetrix expression microarray data of different type of CD4+ T cells, and found that NFκB, JAK-STAT3, TGFβ, and Notch pathway activities were up-regulated in immunosuppressive Treg cells, which may improve prediction and monitor response to checkpoint inhibitor therapy. The findings are clinic relevant, but lack of novelty. Overall, the conclusion is generally supported by the data provided and the manuscript is considered to be suitable for Cancers.

Minor point:

1. The abstract part is better to be polished.

2. Figures 1,2 and 4 mostly like tables. The reviewer prefers table.

3. During the pathway analysis, why the authors only focus on those few pathways, did they include the MAPK and/or other pathways?

Reviewer 3 Report

The authors use Affymetrix expression microarrays data to describe a link between cancer-induced immunotolerance and the type of CD4+ T cells in blood, lymph node, and tumor infiltrate samples derived from patients. They use previously available data as well as their own experimental data to compare the expression profiles of different key signaling pathways in CD4+ T cells. They find correlations that they interpret as an increase in immuno-suppressed CD4+ T cell derived cells - Treg cells - in response to either addition of cancer tissue supernatant in vitro or in patient derived samples. While this study is of significance to identify the signaling pathways that are triggered or suppressed in response to cancer progression to evade immune response, there are some major issues with the results and experimental design that the authors need to address.

A major concern is the low sample size, particularly for patient derived data. The authors need to expand their study to include more patient samples to determine if the differences they observe are significant. Even within the current sample size of 10 patients, the authors observe significant variations in the pathway activities that they attribute to differences in the types of breast cancer. However, without controlling for this, it is unclear if the differences observed are significant and will hold true at a larger sample size. At the very least, the authors need to include a discussion of the limitations of their current study's sample size.

In vitro, the authors observe significant differences between their replicates, however, the cause of these differences is unclear and needs to be investigated. The authors need to repeat their in vitro experiment to activate CD4+ T cells and confirm activation using other methods (flow cytometry, immunofluorescence, western blotting etc.).

The effect of supernatant from cancer tissue on immune suppression is interesting. While the authors include the control of healthy vs. tumor cells, the authors also need to show that this effect is specific to cancer tissue derived supernatant and not in general response to any healthy or apoptotic tissue derived supernatant. This is a minor concern but this would help understand if there are factors specific to cancer tissue that trigger Treg like behaviour in patient derived T-cells.